# Prognostic impact of depressive symptoms on all-cause mortality in individuals with abdominal aortic aneurysm and in the general population: a population-based prospective HUNT study in Norway

Linn Å Nyrønning,[1,2] Rebecka Hultgren ![ORCID],[3,4] Grethe Albrektsen,[5,6] Erney Mattsson,[1,2] Malin Stenman ![ORCID] [4,7]

For numbered affiliations see end of article.

**Correspondence to**
Dr Malin Stenman;
malin.stenman@ki.se

## ABSTRACT

**Background** Abdominal aortic aneurysm (AAA) is a potentially life-threatening disease but the high mortality rate is linked to high age and comorbidity pattern. Depression is associated with increased mortality in the general population and individuals with cardiovascular diseases, but this is sparsely studied for AAA. The aim was to examine the prognostic impact of depressive symptoms on all-cause mortality in individuals with AAA and compare with findings in a general population of the same age and risk profile.

**Methods** Population-based prospective study including 36 616 participants (52.1% women) from the Trøndelag Health Study in Norway. A total of 9428 individuals died during a median follow-up of 10 years at ages 60–90 years. Depressive symptoms were defined by a Hospital Anxiety and Depression Scale-Depression score ≥8. Data on AAA diagnoses and death were obtained from medical records and national registers. HRs from Cox proportional hazard regression models are reported.

**Results** A total of 4832 (13.2%) individuals reported depressive symptoms, whereas 583 (1.6%) AAAs were identified. The adjusted hazard of death was 2.66 times higher in persons with AAA compared with the general population (95% CI 2.39 to 2.97). Overall, there was no significant adverse effect of depressive symptoms in individuals with AAA (HR 1.15;95% CI 0.88 to 1.51), whereas an increased risk was seen in the general population (HR 1.23;95% CI 1.17 to 1.30).

**Conclusion** The overall risk of death was considerably higher in individuals with AAA compared with a general population of the same age and risk profile. Depressive symptoms did not significantly influence the risk of death in the AAA group.

## INTRODUCTION

Abdominal aortic aneurysm (AAA) is a localised widening of the abdominal aorta that often is asymptomatic and undiagnosed.[1] The most feared complication of an AAA is

### Strengths and limitations of this study

► The results from this study are based on data from a large and population-based prospective study in Norway.

► The general population served as comparison group when evaluating the prognostic impact of depressive symptoms on all-cause mortality in individuals diagnosed with abdominal aortic aneurysm (AAA).

► The risk estimates were adjusted for potential confounders, such as age, sex, smoking, coronary heart disease, diabetes, body mass index, hypertension and civil status.

► All AAA cases were verified based on a certified hospital diagnosis, but due to the lack of ultrasound measurement in Norwegian Trøndelag Health Study, the prevalence of AAA in the general population may have been underestimated.

► Data on depressive symptoms and comorbidity patterns were self-reported.

rupture, a life-threatening bleeding associated with fatal outcome if untreated.[1] To prevent deaths from a ruptured AAA, an early diagnosis and treatment in due time is essential. AAA is in general a late onset disease, that is associated with an unfavourable comorbidity pattern and high mortality rates.[2 3] Screening of men aged 65 years or older seems to reduce the risk of deaths related to AAA by up to 30%–40%,[4–7] whereas the effect on all-cause mortality is less clear.[8] Nonetheless, mortality rates in patients that have undergone surgery for AAA have been found to be higher than for patients with other cardiovascular diseases (CVDs).[9 10] High mortality in patients diagnosed with AAA has been linked to high age and unfavourable comorbidity patters,

BMJ

including death from CVDs and other smoking-related diseases. Many individuals diagnosed with AAA live with an asymptomatic aneurysm or undergo elective surgery, and eventually die from other causes.[11] Very few previous studies have compared the overall mortality in individuals diagnosed with AAA to a general population of the same age and with the same risk profile.

Depression is associated with increased mortality in the general population.[12] Moreover, depression is common in patients with CVD[13] and is independently associated with increased cardiovascular morbidity and mortality.[2 14 15] In two recent large prospective, population-based studies, depression was associated with increased risk of AAA.[16 17] AAA and depression have both been associated with low-grade inflammation.[12 18] Hence, inflammation could be a common risk factor that may explain the observed association between AAA and depression. Alternatively, the inflammation process evolves on the pathway between depression and the risk of AAA. Previous studies have suggested that inflammation may increase the risk of rupture by triggering a more rapid growth of the aneurysm.[19 20] If depression, as a proxy for low grade inflammation, influences both the risk and progression of AAA development, an adverse prognostic impact of depression may be particularly pronounced in individuals with AAA. An adverse effect of depression on mortality rates in patients with other CVDs has been established,[12 15 18 21–23] but few previous studies have examined the prognostic impact of depression in individuals with AAA. Moreover, we are not aware of any previous studies that have compared with findings in a general population.

The main aim of this study was to examine whether the prognostic impact of depressive symptoms on all-cause mortality in individuals diagnosed with AAA differed from the established adverse effect in the general population. Moreover, we investigate whether there was an overall difference in all-cause mortality between individuals diagnosed with AAA and the general population of the same age and with the same risk profile.

## METHODS

This is a prospective population-based cohort study, based on data obtained from the Norwegian Trøndelag Health Study (HUNT). The study was performed following the Strengthening the Reporting of Observational Studies in Epidemiology checklist for cohort studies (online supplemental material).

### The HUNT study

HUNT is a large and ongoing multiphase population-based health study of people aged 20 years or older conducted in the county of Nord-Trøndelag in central Norway.[24] The databank contains extensive information about a wide variety of demographics and health-related topics. Data on depressive symptom was introduced in the second HUNT survey (HUNT2).

### Study population

This study population includes 36 616 individuals (52.1 % women) who participated in HUNT2 and/or HUNT3 and had available data in HUNT on depressive symptoms. Median age at the first participation in HUNT was 58.6 years (range 40.6–89.8 years). The unique personal identification number was used to link data on AAA diagnoses (International Statistical Classification of Diseases and Related Health Problems codes 9 and 10) and death with data from the HUNT study.

### Patient and public involvement

This study has no direct contact with patients or the public. The study uses deidentified data from the HUNT study.

### Primary outcome and follow-up period

The primary outcome in this study was all-cause mortality. The Norwegian Cause of Death Registry was used to acquire information on dates of death. Each individual was considered to be at risk of death from the age of 60 years, or from the date of first HUNT participation at a higher age, until date of death, reaching the age of 90 years, or closing date of the study (31 December 2014). The lower and upper age limits were defined due to few deaths before the age of 60 years and few individuals with AAA at risk above the age of 90 years. Median age at start of follow-up was 60 years (range 60.0–89.8 years) and median follow-up time was 10 years (1.1 days to 20 years). At the end of the study period, the median age was 73 years (range 60–90 years). Among the 36 616 individuals included in the study (51.2 % women), 9428 (44.0 % women) died during follow-up.

### Ascertainment of AAA

Information on the diagnosis of AAA in the period 1995 through 2014 was obtained from local hospital registers (in-hospital and outpatient). An indicator variable for whether or not a person had been diagnosed with AAA was defined according to codes 441.3–4 and I71.3–4 in the ninth and tenth editions of the International Classification of Diseases (ICD 9/10 codes). All AAA diagnoses were manually verified using patient charts.

### Assessment of depressive symptoms

The Hospital Anxiety and Depression Scale (HADS) was used to assess the degree of depressive symptoms. HADS is a self-report questionnaire developed to evaluate patients in somatic care and aims to measure levels of anxiety and depression symptoms during the preceding week. The questionnaire consists of two separate subscales ranging from 0 to 21: the HADS Anxiety and HADS Depression (HADS-D). Only the HADS-D score was used in this study. A HADS-D score ≥8 was defined as having depressive symptoms. A cut-off at ≥8 (HADS-D ≥8) gives a specificity of 0.79 and a sensitivity of 0.83 compared with a clinical evaluation of depression.[25 26]

## Demographic and clinical factors

Information on potential confounders was assessed through responses on self-reported questionnaires and clinical examinations. Sex and civil status (categorised as living alone or with other) were assessed through questionnaires. Hypertension (yes, no) was defined as systolic blood pressure >140 mm Hg, diastolic blood pressure >90 mm Hg (average of second and third measurements) or use of antihypertensive medication (self-reported). Body mass index (BMI) was calculated as weight/height$^2$ (kg/m$^2$), and divided into four categories (<25, 25–30, 30–35 and ≥35 kg/m$^2$). Coronary heart disease (CHD, yes, no) was defined as a composite variable of self-reported history of myocardial infarction or angina pectoris. Diabetes (yes, no), smoking status (never, past, current) were defined according to questionnaire responses.

## Statistical analysis

Descriptive statistics at start of follow-up are presented as mean (SD), median (IQR) and/or as frequencies (%), according to nature of the data. Survival analyses with attained age as time scale in the follow-up period were carried out to evaluate the prognostic impact of HADS-D and AAA. The Kaplan-Meier approach was applied to calculate the expected proportion of non-survivors (proportion of deaths) at different ages for AAA (yes, no), and for combined categories of the HADS-D score (<8, ≥8) and the indicator variable for AAA (yes, no). The log-rank test was applied to compare the overall survival between the groups. In addition, the Nelson-Aalen estimate of the cumulative hazard function is presented. The shape of the curves in these plots is informative with respect to changes in hazard with increasing age in each exposure group. Age-adjusted and multivariable adjusted HR for death with 95% CI were calculated as estimate of relative risk of death in a Cox proportional hazard (PH) regression model. A multiplicative interaction term between HADS-D and AAA was added to evaluate whether the association between depressive symptoms and all-cause mortality differed for persons with and without AAA. HUNT participants without a diagnosis of AAA were considered representative for the general population. HR values adjusted for age only (time-scale), and with additional adjustment for sex, BMI category, smoking, CHD, diabetes mellitus, hypertension and civil status are presented. In the analyses of the overall association between AAA and mortality, we also adjusted for depressive symptoms. A backward stepwise procedure was applied to evaluate the individual contributions from single factors in the change from age-adjusted to multivariable adjusted HR-values, and to search for signs of potential overfitting and/or collinearity problems. The PH assumption for each covariate in the Cox PH regression model was visually evaluated in log-minus-log plots. Online supplemental analyses stratified for adjustment factors that showed signs of deviation from the PH assumption were performed. Predicted survival plots based on the Cox PH regression model are also presented. Individuals with missing values on HADS-D (12.6 %) or other covariates were excluded from the analyses (complete case analysis, n=34 464).

## Time-varying exposure factors

Except for sex, all prognostic factors were treated as time-varying covariates in the Cox PH regression model, allowing a shift in exposure categories with increasing age for individuals who participated in two HUNT surveys. The covariate values at start of follow-up were defined according to values obtained at the HUNT participation closest in time before the age of 60 years or at the first HUNT participation at an older age. In the Kaplan-Meier survival analyses, the 'at risk' category was defined according to the most recent registered value of the exposure factor during follow-up. The indicator variable for AAA was 0 (not present) until date (age) of diagnosis (recoded to 1, that is, present). The diagnosis of AAA could occur either before, after or between two HUNT participations, depending on age at diagnosis and age at participation in a HUNT survey. A total of 27 individuals (2 women) had been diagnosed with AAA at start of follow-up from the age of 60 years, whereas an additional 556 individuals (149 women) were diagnosed during the follow-up.

All statistical analyses were performed using Stata (V.IC.16.1). A p<0.05 was considered significant (two-sided tests).

## RESULTS

The distribution of the demographic and clinical characteristics of the study population at start of follow-up is presented in table 1. A total of 583 (1.6 %) individuals (432 men, 151 women) had a diagnosed AAA, and 4 832 (13.2%) individuals reported depressive symptoms (HADS-D ≥8). The proportion with a HADS-D score ≥8 was higher in individuals with than without a diagnosed AAA (table 1). Compared with persons without AAA (considered representative for the general population), individuals with AAA were more often men (74.1 vs 48.4 %), current smokers (57.5 vs 35.5%) and reported more CHD (27.4 vs 8.9 %) and hypertension (72.5 vs 57.5 %). Moreover, a larger proportion in the AAA group was obese (4.7 vs 3.0 %), but no significant association with BMI category was found.

### All-cause mortality in individuals with AAA and in the general population

A Kaplan-Meier failure plot, with estimated proportion of deaths at different ages, for individuals with and without a diagnosed AAA, and Nelson-Aalen estimates of the cumulative hazard function, are shown in figure 1. Individuals with AAA died at a younger age compared with the general population (log rank test, p<0.001; median age at death 73.0 vs 85.2 years). In the Cox PH regression model adjusted for age only, the hazard of death in individuals with AAA was more than three times the hazard of

**Table 1** Demographic and clinical characteristics at start of follow-up, total and by AAA

| Characteristics | Total population n (%) | AAA n (%) | No AAA n (%) | P value |
|---|---|---|---|---|
| **Total** | 36 616 (100) | 583 (100) | 36 033 (100) | |
| Sex | | | | <0.001 |
| Men | 17 883 (48.8) | 432 (74.1) | 17 451 (48.4) | |
| Women | 18 733 (51.2) | 151 (25.9) | 18 582 (51.6) | |
| Age at measured HADS-D | | | | <0.001 |
| <60 | 20 843 (56.9) | | | |
| 60–69 | 9550 (26.1) | | | |
| 70–79 | 5254 (14.3) | | | |
| ≥80 | 969 (2.4) | | | |
| HADS-D | | | | <0.001 |
| <8 | 31 784 (86.3) | 477 (81.8) | 31 307 (86.9) | |
| ≥8 | 4832 (13.2) | 106 (18.2) | 4726 (13.1) | |
| Smoking | | | | <0.001 |
| Never | 12 890 (35.9) | 53 (9.2) | 12 837 (36.4) | |
| Past | 12 738 (35.5) | 192 (33.5) | 12 546 (35.5) | |
| Current | 10 260 (28.6) | 12 546 (35.5) | 9931 (28.1) | |
| BMI | | | | 0.26 |
| <25 | 11 198 (30.8) | 177 (30.7) | 11 021 (30.8) | |
| 25–29 | 17 349 (47.8) | 280 (48.5) | 17 069 (47.7) | |
| 30–35 | 6107 (16.8) | 103 (17.8) | 6004 (16.8) | |
| ≥35 | 1679 (4.6) | 17 (3.0) | 1662 (4.7) | |
| CHD | 3347 (9.2) | 423 (27.4) | 160 (8.9) | <0.001 |
| Hypertension | 21 124 (57.7) | 423 (72.5) | 20 701 (57.5) | <0.001 |
| Diabetes | 1746 (4.8) | 23 (4.0) | 1723 (4.8) | 0.09 |
| Civil status | | | | 0.35 |
| Cohabiting | 26 650 (72.9) | 443 (76.0) | 26 207 (72.8) | |
| Living alone | 9913 (27.1) | 140 (24.0) | 9773 (27.2) | |

Missing data among those with valid HADS-D (n=36616): Smoking, n=728 (2.6%) BMI, n=283 (0.8%), CHD, n=23 (0.06%), systolic and diastolic blood pressure, n=154 (0.4%), hypertension, n=9 (0.02%), diabetes, n=84 (0.2%), civil status, n=53 (0.14%).
Missing data on HADS-D, n=5259 (12.6%).
AAA, Abdominal Aortic Aneurysm; BMI, body mass index; CHD, coronary heart disease; HADS-D, Hospital Anxiety and Depression Scale-Depression.

death in the general population (table 2, HR 3.39, 95% CI 3.01 to 3.78). The difference was less pronounced in the multivariable analysis, but the hazard remained more than twice as high in the AAA group (table 2, HR 2.66, 95% CI 2.39 to 2.97).

### Prognostic impact of depression in individuals with AAA and in the general population

At all ages, the estimated proportions of deaths were higher among individuals with than without depressive symptoms (HADS-D ≥8 vs >8), both in individuals diagnosed with AAA and in the general population (figure 2). Median age at death in individuals with and without depressive symptoms in the AAA group was 70.8 and 73.6 years, whereas the corresponding numbers in the general population were 83.7 and 85.3 years.

In the age-adjusted Cox PH regression analysis, the adverse effect of depressive symptoms in individuals with AAA was rather similar to the association seen in the general population (table 2, HR for HADS-D ≥8 vs <8 of 1.28 in both groups). In the analyses adjusted for age, sex, smoking, CHD, diabetes, BMI, hypertension and civil status (table 2) the HR value for HADS-D was closer to unity and no longer statistically significant in individuals with AAA (HR 1.15, 95% CI 0.88 to 1.51), whereas an adverse effect remained in the general population (HR 1.23, 95% CI 1.17 to 1.30). The difference in these estimates (1.15 vs 1.23) did not reach statistical significance

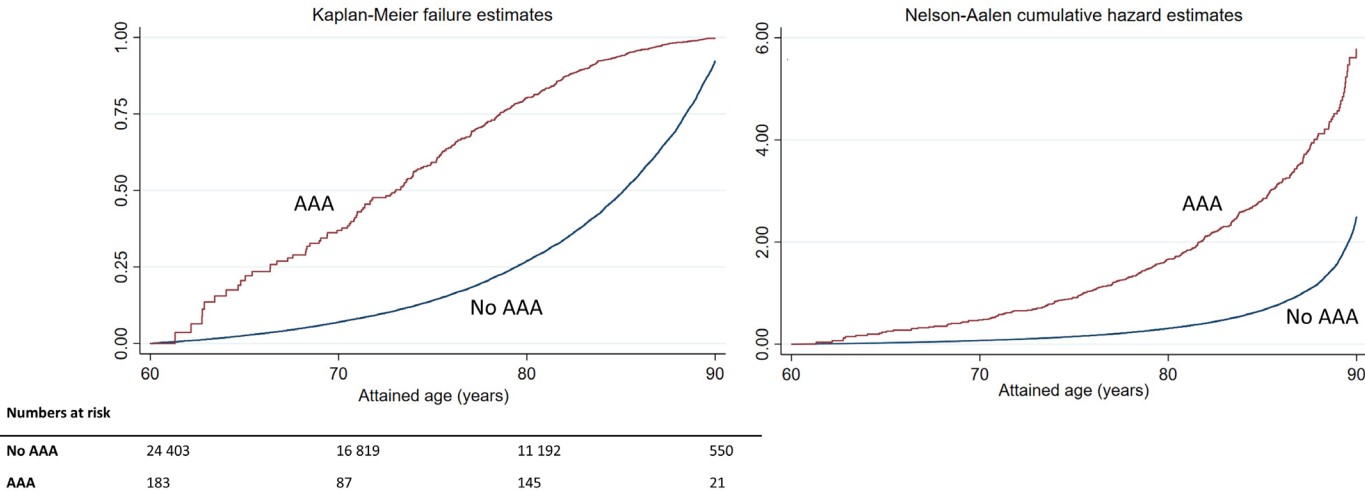

**Figure 1** Kaplan-Meier failure plot and Nelson-Aalen cumulative hazard estimates in subgroups defined by AAA (no, yes). AAA, abdominal aortic aneurysm.

(p, test for interaction=0.62). The adjustment factors that contributed the most to the reduced strength of the association with depressive symptoms were CHD, BMI and smoking. There was no single factor that explained all the flattening of the risk estimate or led to very imprecise estimates.

HR values (multivariable adjusted) for combined categories of HADS-D and AAA, using a common reference group, to allow for a direct comparison of relative risk across all subgroups, are shown in figure 3. The adverse, though weak association with depression appeared to be rather similar in individuals with and without AAA in this analysis. The prognostic impact of AAA was considerably more pronounced than the prognostic impact of depressive symptoms.

## Age-dependent effects

The Kaplan-Meier failure plot, as well as the Nelson-Aalen cumulative hazard plot (figure 2), indicated a more pronounced negative prognostic effect from depressive symptoms in AAA individuals younger than 70 years. No such age-dependent effect of depressive symptoms was seen in the general population (figure 2). The age-dependent pattern was confirmed in the log-log-plot used to evaluate the PH assumption in the Cox regression model (online supplemental figure 1). The age-dependent pattern in the AAA group remained in the analyses adjusted for potential confounders, as illustrated in the predicted survival plots based on a multivariable adjusted Cox-PH regression models stratified on HADS-D

**Table 2** HR with 95% CI of death for AAA (yes vs no) and for depressive symptoms (HADS-D ≥8 vs <8) in subgroups defined by AAA (yes, no)*

|  | No of deahts | Age adjusted HR (95% CI)* | Multivariable HR (95% CI)† |
|---|---|---|---|
| AAA |  |  |  |
| No | 9078 | 1.00 | 1.00 |
| Yes | 350 | 3.39 (3.01 to 3.78) | 2.66 (2.39–2.97) |
| HADS-D |  |  |  |
| Individuals with AAA |  |  |  |
| <8 | 283 | 1.00 | 1.00 |
| ≥8 | 67 | 1.28 (0.98 to 1.69) | 1.15 (0.88–1.51) |
| Individuals without AAA |  |  |  |
| <8 | 7430 | 1.00 | 1.00 |
| ≥8 | 1648 | 1.28 (1.21 to 1.36) | 1.23 (1.17–1.30) |
| P value, interaction |  | 0.90 | 0.62 |

Depressive symptoms were defined as a HADS-D score ≥8.
*Based on Cox PH regression model with attained age as time scale.
†Additional adjustment for sex, coronary heart disease, diabetes mellitus, body mass index-category, smoking, hypertension, civil status. HR for AAA also adjusted for HADS-D. Complete case analyses: Number of individuals, n=34 464.
AAA, abdominal aortic aneurysm; HADS-D, Hospital Anxiety and Depression Scale-Depression; HR, hazard ratio.

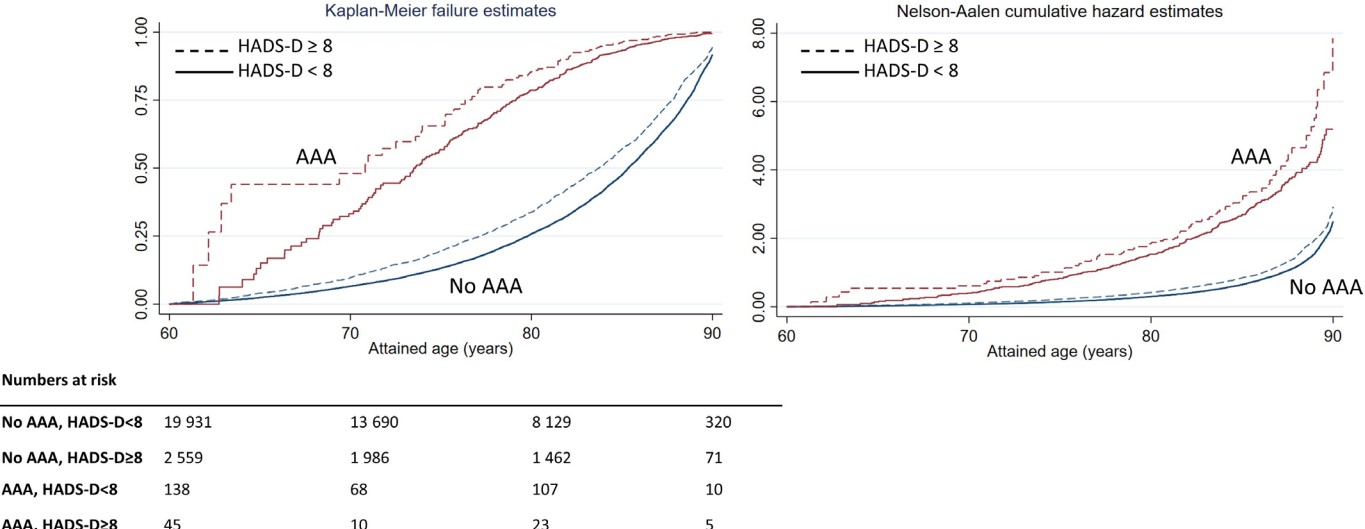

**Figure 2** Kaplan-Meier failure plot and Nelson-Aalen cumulative hazard estimates in subgroups defined by HADS-D (<8 and ≥8) and AAA (no, yes). AAA, abdominal aortic aneurysm; HADS-D, Hospital Anxiety and Depression Scale-Depression.

and AAA to allow for deviation from the PH assumption (figure 4). HR values are not calculated for stratification variables in the Cox PH model, and insufficient amount of data at the youngest ages in certain subgroups, precluded an age-specific analysis. Thus, age-specific HR values for HADS-D (≥8 vs <8) in subgroups defined by AAA (no, yes) cannot be reported.

Log-log plot also indicated deviation from the PH assumption for BMI category. Additional analyses using BMI category as a stratification variable in the Cox regression model did not notably impact the HR values for our main study variables (AAA and HADS-D).

## DISCUSSION

In this large, population-based prospective study, the overall risk of death was more than twice as high in individuals with AAA compared with a general population at the same age and same risk profile. There was no

clear adverse effect of depressive symptoms on all-cause mortality in individuals diagnosed with AAA after adjustment for potential confounders, whereas a negative prognostic effect remained in the general population. At ages <70 years, however, there was a tendency towards a more pronounced adverse effect of depressive symptoms in individuals with AAA. This finding was uncertain due to the low number of deaths before the age of 70 years.

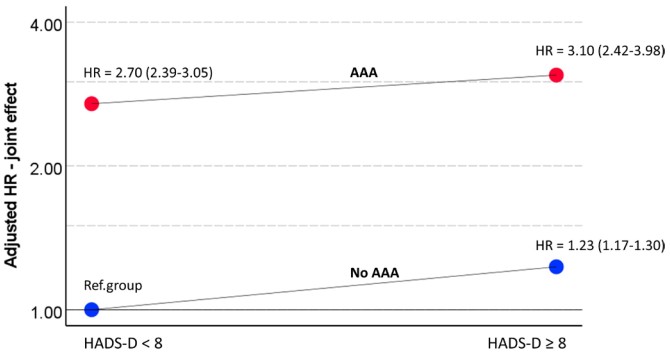

**Figure 3** Hazard ratio with 95% confidence interval for the combined categories of AAA (No, Yes) and HADS-D (<8, ≥8), using a common reference group (no AAA, HADS-D <8), plotted on logarithmic scale (base 2). The HR values are calculated in a multivariable adjusted Cox pH regression model. AAA, Abdominal Aortic Aneurysm, HADS, Hospital Anxiety and Depression Scale.

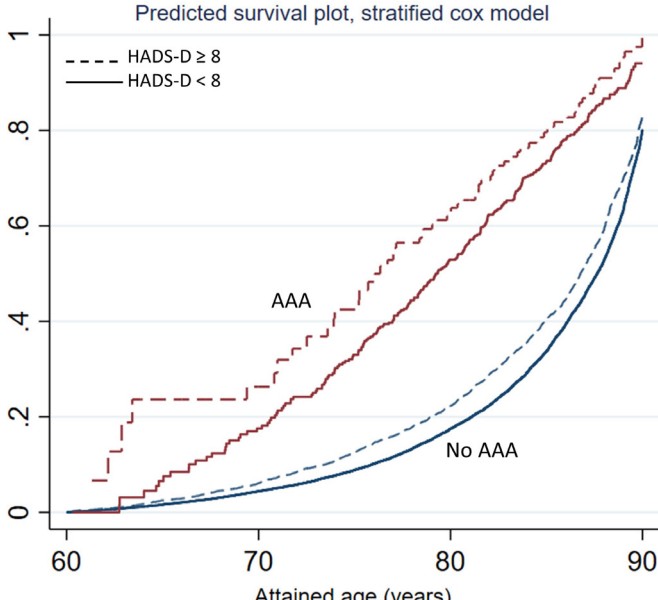

**Figure 4** Predicted survival plots based on a Cox PH regression model stratified for HADS-D and AAA (combined categories) to allow for an age-dependent effect. Curves are calculated for reference categories of adjustment factors (sex, BMI category, smoking, CHD, diabetes mellitus, hypertension and civil status). Complete-case analyses. AAA, abdominal aortic aneurysm; BMI, body mass index; CHD, coronary heart disease; HADS-D, Hospital Anxiety and Depression Scale-Depression; PH, proportional hazard.

The finding of a substantially higher mortality in individuals with vs without AAA is consistent with a recent study reporting markedly lower overall long-term survival in patients undergoing successful AAA repair as compared with age-adjusted and sex-adjusted mortality rates in the general population.[27] An unfavourable comorbidity profile commonly seen in AAA patients, including other CVDs, as well as smoking, is probably a contributing factor to the high crude mortality rates in individuals with AAA.[28] In our study, however, the relative risk estimate remained high also after adjusting for factors associated with both AAA and increased mortality, such as sex, CHD, diabetes and smoking. We are not aware of any previous studies that have compared mortality in individuals with AAA with a general population of the same age and also same level of other risk factors. Most previous studies have almost exclusively been restricted to patients that have undergone surgery for AAA or patients deemed unfit for surgery.[27 29–31] This study included both surgically untreated and treated AAA patients, but we did not distinguish between these subgroups in the analyses due to lack of information on this issue.

To our knowledge, no previous studies have examined whether depressive symptoms on mortality differ in patients with AAA compared with a general population. The results from the this study did not confirm our initial hypothesis that a negative prognostic impact of depressive symptoms may be stronger in individuals with AAA compared with a general population. On the contrary, there was no clear association between depressive symptoms and mortality in the AAA group. The lack of association could relate to the low number of individuals with both AAA and depressive symptoms. However, we did observe a slightly more pronounced adverse effect of depressive symptoms in individuals diagnosed with AAA before the age of 70 years, whereas no such age-dependent effect was seen in the general population. This pattern may relate to the fact that some risk factors for AAA, such as a high number of pack years (smoking) and unfavourable heredity patterns have been reported to be more common in young compared with older AAA patients.[32] We did adjust for smoking habits, whereas data on heritability was not available. Due to the low number of individuals in these age groups, these results must be interpreted with caution and confirmed in other studies.

The biological mechanisms behind the increased mortality rates in individuals with AAA are yet to be clarified. Inflammation seems to be an important part of both the AAA pathogenesis and growth,[33 34] which may relate to impact from noxious stimuli such as cigarette smoke, low density lipoprotein cholesterol and hypertension.[35] With increasing age, competent repair cells are not always available, artery repair fails and the inflammatory state increases.[1] Previous studies have shown that depression is associated with low-grade chronic inflammation.[1 15 36] Moreover, a negative impact from inflammatory processes on mortality in patients with CVDs and concurrent depression has been suggested.[37] Our finding indicating

that depressive symptoms might worsen the prognosis in younger, but not in older individuals diagnosed with AAA might be somewhat contradictory. On the other hand, it is possible that individuals who develop AAA at a younger age have faster aneurysm growth that is more susceptible to a growth-enhancing effect from depression, potentially mediated by inflammation.

### Strengths and limitations

A major strength of our study is that the results are based on data from a large, population-based, prospective study. The HUNT population is a validated and well-defined homogeneous cohort that represents the general population in Norway. Data on risk factors were assessed at the individual level for all study participants, enabling tight adjustment for age (1- year intervals, time scale in the survival analyses), sex and other potential confounders when estimating HR values. Repeated measurements of the exposure factors made it possible to update the risk set by use of time-varying covariates. This analytic approach reduces potential misclassification bias in the risk estimates. However, not all the individuals included in our study had participated in HUNT more than once, which is a limitation. The amount of missing data for the covariates was low. All AAA cases were verified based on a certified hospital diagnosis and dates of death were obtained from national registers with compulsory recordings. It is, however, likely that there was some underdiagnosis of AAA cases. The HUNT study does not include ultrasound examinations of the abdominal aorta. The observed 1.6% crude prevalence of AAA in our study is, however, in line with the expected rate at the time.[38] Unless a potential underdiagnosis of AAA differ by value of HADS-D, the risk estimates will not be biased. Some limitations must be considered. Most data on covariates were self-reported, including symptoms of depression, cardiovascular comorbidities and smoking, potentially leading to information bias. Symptoms of depression were assessed through self-reported questionnaires and not by a psychiatric diagnostic interview. We had no information about antidepressant use. The low number of individuals with severe depressive symptoms in our study population could be a result from recruitment bias. Individuals with severe depression might not adhere to screening programmes or attend population-based studies to the same extent as non-depressed individuals. However, such recruitment bias would likely bias our results towards the null hypothesis.

### CONCLUSION

In this population-based prospective study, there was no significant prognostic impact of depressive symptoms on mortality in individuals with AAA, whereas an increased risk was seen in the general population. Our results revealed a hitherto unreported higher all-cause mortality in individuals with AAA compared with a general population of the same age and same risk profile. More studies

with larger AAA cohorts are needed to explore the relation between depression, AAA and mortality and a potential heterogeneity across age groups.

**Author affiliations**
[1]Department of Surgery, Vascular Surgery, St Olav University Hospital, Trondheim, Norway
[2]Department of Circulation and Medical Imaging, NTNU, Trondheim, Norway
[3]Department of Vascular Surgery, Karolinska University Hospital, Stockholm, Sweden
[4]Department of Molecular Medicine and Surgery, Karolinska Institutet, Stockholm, Sweden
[5]Department of Public Health and Nursing, NTNU, Trondheim, Norway
[6]Department of Clinical and Molecular Medicine, NTNU, Trondheim, Norway
[7]Perioperative Medicine and Intensive Care function, Karolinska University Hospital, Stockholm, Sweden

**Acknowledgements** The Trøndelag Health Study (The HUNT Study) is a collaboration between HUNT Research Centre, (Faculty of Medicine and Health Sciences, NTNU, Norwegian University of Science and Technology), Trøndelag County Council, Central Norway Regional Health Authority and the Norwegian Institute of Public Health.

**Contributors** LÅN contributed to study conception and design, data and statistical analysis, interpretation of study findings and drafting and critical review of the manuscript and is also responsible for the overall content as guarantor. RH contributed to study conception and design, interpretation of study findings and drafting and critical review of the manuscript. GA contributed to study conception, and statistical methods and analysis, interpretation of study findings and critical review of the manuscript. EM contributed to study conception and design and critical review of the manuscript. MS contributed to study conception and design, interpretation of study findings and drafting and critical review of the manuscript.

**Funding** This research has been supported by the Swedish Heart-Lung Foundation (Hultgren) grant number: 20190553 and 20180506.

**Competing interests** None declared.

**Patient consent for publication** Not applicable.

**Ethics approval** The study has been approved by the HUNT Study board of directors and the Regional Ethics Committee (2014/175/REK12 Midt-Norge).

**Provenance and peer review** Not commissioned; externally peer reviewed.

**Data availability statement** No data are available. The Trøndelag Health Study (HUNT) has invited persons aged 13–100 years to four surveys between 1984 and 2019. Comprehensive data from more than 140 000 persons having participated at least once and biological material from 78 000 persons are collected. The data are stored in HUNT databank and biological material in HUNT biobank. HUNT Research Centre has permission from the Norwegian Data Inspectorate to store and handle these data. The key identification in the data base is the personal identification number given to all Norwegians at birth or immigration, whilst deidentified data are sent to researchers upon approval of a research protocol by the Regional Ethical Committee and HUNT Research Centre. To protect participants' privacy, HUNT Research Centre aims to limit storage of data outside HUNT databank, and cannot deposit data in open repositories. HUNT databank has precise information on all data exported to different projects and are able to reproduce these on request. There are no restrictions regarding data export given approval of applications to HUNT Research Centre. For more information see: https://www.ntnu.edu/hunt/data.

**ORCID iDs**
Rebecka Hultgren http://orcid.org/0000-0002-8869-0493
Malin Stenman http://orcid.org/0000-0002-6782-0586

1 Sakalihasan N, Michel J-B, Katsargyris A, et al. Abdominal aortic aneurysms. *Nat Rev Dis Primers* 2018;4:34.
2 Hare DL, Toukhsati SR, Johansson P, et al. Depression and cardiovascular disease: a clinical review. *Eur Heart J* 2014;35:1365–72.
3 Bath MF, Sidloff D, Saratzis A, et al. Impact of abdominal aortic aneurysm screening on quality of life. *Br J Surg* 2018;105:203–8.
4 Wanhainen A, Hultgren R, Linné A, et al. Outcome of the Swedish nationwide abdominal aortic aneurysm screening program. *Circulation* 2016;134:1141–8.
5 Norman PE, Jamrozik K, Lawrence-Brown MM, et al. Population based randomised controlled trial on impact of screening on mortality from abdominal aortic aneurysm. *BMJ* 2004;329:1259.
6 Ashton HA, Buxton MJ, Day NE, et al. The multicentre aneurysm screening study (mass) into the effect of abdominal aortic aneurysm screening on mortality in men: a randomised controlled trial. *Lancet* 2002;360:1531–9.
7 Lindholt JS, Juul S, Fasting H, et al. Screening for abdominal aortic aneurysms: single centre randomised controlled trial. *BMJ* 2005;330:750.
8 Ali MU, Fitzpatrick-Lewis D, Miller J, et al. Screening for abdominal aortic aneurysm in asymptomatic adults. *J Vasc Surg* 2016;64:1855–68.
9 Timmers TK, van Herwaarden JA, de Borst G-J, et al. Long-Term survival and quality of life after open abdominal aortic aneurysm repair. *World J Surg* 2013;37:2957–64.
10 de Bruin JL, Baas AF, Heymans MW, et al. Statin therapy is associated with improved survival after endovascular and open aneurysm repair. *J Vasc Surg* 2014;59:39–44.
11 Bath MF, Gokani VJ, Sidloff DA, et al. Systematic review of cardiovascular disease and cardiovascular death in patients with a small abdominal aortic aneurysm. *Br J Surg* 2015;102:866–72.
12 Mykletun A, Bjerkeset O, Overland S, et al. Levels of anxiety and depression as predictors of mortality: the HUNT study. *Br J Psychiatry* 2009;195:118–25.
13 Harshfield EL, Pennells L, Schwartz JE, et al. Association between depressive symptoms and incident cardiovascular diseases. *JAMA* 2020;324:2396–405.
14 Stenman M, Holzmann MJ, Sartipy U. Relation of major depression to survival after coronary artery bypass grafting. *Am J Cardiol* 2014;114:698–703.
15 Nemeroff CB, Goldschmidt-Clermont PJ. Heartache and heartbreak--the link between depression and cardiovascular disease. *Nat Rev Cardiol* 2012;9:526–39.
16 Daskalopoulou M, George J, Walters K, et al. Depression as a risk factor for the initial presentation of twelve cardiac, cerebrovascular, and peripheral arterial diseases: data linkage study of 1.9 million women and men. *PLoS One* 2016;11:e0153838.
17 Nyrønning Linn Åldstedt, Stenman M, Hultgren R, et al. Symptoms of Depression and Risk of Abdominal Aortic Aneurysm: A HUNT Study. *J Am Heart Assoc* 2019;8:e012535.
18 Pan A, Sun Q, Okereke OI, et al. Depression and risk of stroke morbidity and mortality: a meta-analysis and systematic review. *JAMA* 2011;306:1241–9.
19 Sakalihasan N, Van Damme H, Gomez P, et al. Positron emission tomography (PET) evaluation of abdominal aortic aneurysm (AAA). *Eur J Vasc Endovasc Surg* 2002;23:431–6.
20 Dale MA, Ruhlman MK, Baxter BT. Inflammatory cell phenotypes in AAAS: their role and potential as targets for therapy. *Arterioscler Thromb Vasc Biol* 2015;35:1746–55.
21 Almeida OP, Ford AH, Hankey GJ, et al. Depression, antidepressants and the risk of cardiovascular events and death in older men. *Maturitas* 2019;128:4–9.
22 Cuijpers P, Vogelzangs N, Twisk J, et al. Comprehensive meta-analysis of excess mortality in depression in the general community versus patients with specific illnesses. *Am J Psychiatry* 2014;171:453–62.
23 Letnes JM, Torske MO, Hilt B, et al. Symptoms of depression and all-cause mortality in farmers, a cohort study: the HUNT study, Norway. *BMJ Open* 2016;6:e010783.
24 Krokstad S, Langhammer A, Hveem K, et al. Cohort profile: the HUNT study, Norway. *Int J Epidemiol* 2013;42:968–77.
25 Bjelland I, Dahl AA, Haug TT, et al. The validity of the hospital anxiety and depression scale. An updated literature review. *J Psychosom Res* 2002;52:69–77.

26 Myrtveit SM, Ariansen AMS, Wilhelmsen I, *et al*. A population based validation study of self-reported pensions and benefits: the Nord-Trøndelag health study (Hunt). *BMC Res Notes* 2013;6:27.

27 Bulder RMA, Talvitie M, Bastiaannet E, *et al*. Long-Term prognosis after elective abdominal aortic aneurysm repair is poor in women and men: the challenges remain. *Ann Surg* 2020;272:773–8.

28 Toghill BJ, Saratzis A, Bown MJ. Abdominal aortic aneurysm-an independent disease to atherosclerosis? *Cardiovasc Pathol* 2017;27:71–5.

29 Whittaker JD, Meecham L, Summerour V, *et al*. Outcome after Turndown for elective abdominal aortic aneurysm surgery. *Eur J Vasc Endovasc Surg* 2017;54:579–86.

30 Rueda-Ochoa OL, van Bakel P, Hoeks SE, *et al*. Survival after uncomplicated EVAR in octogenarians is similar to the general population of octogenarians without an abdominal aortic aneurysm. *Eur J Vasc Endovasc Surg* 2020;59:740–7.

31 Norman PE, Semmens JB, Lawrence-Brown MM, *et al*. Long term relative survival after surgery for abdominal aortic aneurysm in Western Australia: population based study. *BMJ* 1998;317:852–6.

32 Howard DPJ, Banerjee A, Fairhead JF, *et al*. Age-Specific incidence, risk factors and outcome of acute abdominal aortic aneurysms in a defined population. *Br J Surg* 2015;102:907–15.

33 Kuivaniemi H, Ryer EJ, Elmore JR, *et al*. Understanding the pathogenesis of abdominal aortic aneurysms. *Expert Rev Cardiovasc Ther* 2015;13:975–87.

34 Eagleton MJ. Inflammation in abdominal aortic aneurysms: cellular infiltrate and cytokine profiles. *Vascular* 2012;20:278–83.

35 Rocha VZ, Libby P, Obesity LP. Obesity, inflammation, and atherosclerosis. *Nat Rev Cardiol* 2009;6:399–409.

36 Raedler TJ. Inflammatory mechanisms in major depressive disorder. *Curr Opin Psychiatry* 2011;24:519–25.

37 Celano CM, Huffman JC. Depression and cardiac disease: a review. *Cardiol Rev* 2011;19:130–42.

38 Ulug P, Powell JT, Sweeting MJ, *et al*. Meta-Analysis of the current prevalence of screen-detected abdominal aortic aneurysm in women. *Br J Surg* 2016;103:1097–104.

