## [Reviewer comments · BMJ Open]

ARTICLE DETAILS

TITLE (PROVISIONAL)	The prognostic impact of depressive symptoms on all-cause mortality in individuals with abdominal aortic aneurysm and in the general population. A population-based prospective HUNT study in Norway
AUTHORS	Nyrønning, Linn Å.; Hultgren, Rebecka; Albrektsen, Grethe; Mattsson, Erney; Stenman, Malin

VERSION 1 – REVIEW

REVIEWER	Udumyan, Ruzan Örebro Universitet, Clinical Epidemiology and Biostatistics, School of Medical Sciences
REVIEW RETURNED	02-Mar-2021

GENERAL COMMENTS	Review for the paper: Mortality in individuals with Abdominal Aortic Aneurysm Do depressive symptoms worsen the prognosis? A HUNT Study Comments to the authors Title: The title looks like being composed of three disconnected segments. Also, it suggests that the analysis is performed among patients with AAA. Please revisit. Any reasons for writing Abdominal Aortic Aneurysm with capital letters? Are abbreviations allowed in the title? Abstract: The background is kind of disconnected from the aim. The provided background information “Depression is associated with increased mortality in individuals with cardiovascular diseases, but such association is sparsely studied for AAA.” may imply that you aimed to study the association between depression and mortality among patients with AAA. However, in the aims you speak about individuals with and without AAA. The aim is not clearly formulated. If you view AAA as an exposure and depression as a potential effect modifier (as may look from the title), then please formulate the aim accordingly. “However, younger AAA patients (< 70 years) reporting depressive symptoms appeared to have a particularly poor prognosis.” Please provide HRs. Strengths and limitations of this study
--

Page 4, line 23: "Adjustment were made for age, sex and known comorbidities." – what is meant by "known comorbidities"? - comorbidity that are associated with all-cause mortality?

Page 4: lines 8-10: "This is the first study to investigate the association between depressive symptoms and mortality in patients with AAA compared to a general population" vs lines 34-38: "The study focused on the association between depressive symptoms and mortality in patients with AAA." do not test the same hypothesis and imply different analytic strategies.

Introduction:

Pages 5-6: "If depression influences the risk of AAA development, AAA patients suffering from depression may have higher mortality rates compared to AAA patients without depression." The statement implies that if depression is associated with AAA development, then depression may act as an effect modifier in the AAA-mortality association. However, confounders are also associated with the exposure of interest by definition- so, the statement is inappropriate. Please rephrase and clarify the hypothesized interrelationships and pathways. Perhaps, you hypothesized that depression may act as an effect modifier in the AAA-mortality association through its association with inflammation?

Statistical analysis:

Page 9: "An interaction term between HADS and AAA was included in the model to evaluate whether the prognostic impact of depressive symptoms differed between persons with and without AAA." Please specify that you mean multiplicative interaction term. Suggest you test both multiplicative and additive interactions. Also, some statements imply that your exposure of interest is AAA diagnosis, whereas depression is a potential effect modifier, while the sentence above implies that depression is your exposure variable of interest. Please be consistent and clear.

Pages 9-10: "Predicted survival plots based on an adjusted Cox PH regression model, stratified for AAA and depressive symptoms, were constructed to allow for deviation from the PH assumption." How did you allow for deviation from PH assumption if the model was a PH model? Please note difference between terms 'time-varying covariate' and 'time-dependent effect'.

Page 10: "Attained age was used as time scale both in the Kaplan-Meier survival analyses and in the Cox PH regression model." Survival function is hard to interpret when age is used as a time-scale and hence not recommended. You could plot hazard function instead as it remains interpretable in terms of the risk experienced by people of a given age.

Results:

Page 11: provide a brief summary of descriptive statistics included in Table 1.

Discussion:

Page 15: "Previously reported mortality rates in AAA patients have most often been restricted to patients treated for AAA. [26, 28] In contrast, our study included both untreated and treated AAA patients." Could you adjust for the treatment?

REVIEWER	Martin, Guy Imperial College London, Department of Surgery & Cancer
REVIEW RETURNED	08-Mar-2021

GENERAL COMMENTS	General Comments: This prospective population-based study sought to identify the impact of depressive symptoms on mortality in patients with AAA. A sub-group of AAA patients (583, 1.6%) was identified from a larger study group. On the whole it is a well written and constructed analysis from an established longitudinal study population with good data quality and completeness. The approach to analysis is well understood and has been extensively published by the study group in the past. The discussion and explanation of the results in the context of the wider literature is particularly well constructed. Generally, in the results, I find the narrative a little lost. The narrative is generally arguing that depression in AAA leads to a worse prognosis, yet the key result in Table 2 is that depression in those with AAA has no impact on mortality (HR 1.12, 0.88-1.51). I think the overall messaging of the result section does not fit with this and therefore should be revised to less strongly support the link between depression and outcomes in AAA. In fact, this is neatly highlighted in the discussion which is well written and framed; the results presented and how they are described in both the abstract and main text do not match that which is later discussed. Broadly speaking the overall message here is that of the single result that depression does not appear impact mortality in AAA that must be presented within the limitations of the statistical analysis. The remainder of the results presented merely demonstrate what is already well established in the literature (but in a relatively small sample size here) that AAA patients have a generally higher mortality risk, that most likely related to excess cardiovascular disease risks and smoking. Specific Comments: Abstract - The results/conclusions presented in the abstract do not seem coherent. Increased mortality in those with a AAA diagnosis compared to those without in the general population is expected and is not a novel or relevant finding. No data is provided for the size or significance for the effect of depressive symptoms in young patients with AAA despite stating they have a “particularly poor prognosis”. Essentially, the only conclusion that can really be drawn from the results provided is that depressive symptoms have no impact on mortality in patients with AAA. The results/conclusions presented require further clarity based on this. P5L10 - arguably it is almost certainly 100% mortality, but some people do survive untreated ruptures in the short term.
---

	P5L45 - I wonder whether depression is a true independent risk-factor for AAA, or in fact is confounded by its association to smoking which is also a highly relevant risk factor for AAA. This was addressed in the original references provided which caveat the limitations of the statistical models used in them, in particular smoking being treated as a time-dependent variable. I think caveats such as this need to be addressed in framing the underlying hypothesis presented linking AAA mortality to depression. Is it also that depressed people are less likely to engage with medical services and therefore not attend AAA screening? This should also be highlighted. P9L20 - there is no discussion of how the risk of collinearity of data has been assessed and mitigated (e.g., smoking with depression), and how this together with the number of AAA events may lead to over-fitting of the model. This should be discussed/accounted for. P11L20 - I think the key results discussed here should be presented in the text, rather than just referenced in a table (e.g., the adjusted HR for death in the AAA group, impact of younger age on prognostic impact of AAA). This should be done throughout the results section in order to make it easier for the reader to follow and understand. P16L47 - the authors are correct to mention the potential risk of heritability on AAA in young patients. It would be sensible to specifically reference the connective tissue disorder cohort here who may be responsible for the weak association seen, but for whom data is not available. Figure 2A - the key problem with this figure, and therefore the relevance/significance of the findings is the small numbers at risk, particularly for patients with a AAA and HADS >8. Arguably the numbers are too small to make the plot justified.
--	--

VERSION 1 – AUTHOR RESPONSE

Response to Reviewers

Reviewer: 1

Dr. Ruzan Udumyan, Örebro Universitet

Comments to the Author:

Review for the paper:

Mortality in individuals with Abdominal Aortic Aneurysm

Do depressive symptoms worsen the prognosis? A HUNT Study

Comments to the authors

Title:

The title looks like being composed of three disconnected segments. Also, it suggests that the analysis is performed among patients with AAA. Please revisit.

Any reasons for writing Abdominal Aortic Aneurysm with capital letters?

Are abbreviations allowed in the title?

Thank you, we agree that it could improve in clarity. It is now revised.

Abstract:

The background is kind of disconnected from the aim. The provided background information “Depression is associated with increased mortality in individuals with cardiovascular diseases, but such association is sparsely studied for AAA.” may imply that you aimed to study the association between depression and mortality among patients with AAA. However, in the aims you speak about individuals with and without AAA.

The aim is not clearly formulated. If you view AAA as an exposure and depression as a potential effect modifier (as may look from the title), then please formulate the aim accordingly.

Thank you for pointing this out. We agree with the reviewer’s comment regarding the formulation of the aim. The introduction in the abstract has been reformulated, and the aim has been clarified.

“However, younger AAA patients (< 70 years) reporting depressive symptoms appeared to have a particularly poor prognosis.” Please provide HRs.

Thank you for the comment. We are not able to report HRs regarding this issue. We noticed a potential age-specific impact from depressive symptoms in the youngest age group, as illustrated in the survival plot based on the Kaplan-Meier method. This method does not provide a HR value. However, the same pattern was seen in predicted survival plots based on an adjusted and stratified cox ph model but HR values are not calculated for the stratification variable in the Cox model. A low number of deaths and few individuals with both AAA and depressive symptoms in the age group below 70 years, precluded further age-specific analyses. We acknowledge that this is a limitation of our study. We have removed the above sentence from the abstract, and have also changed the section regarding this issue in the results (page 12-13 line/line 17-4).

Strengths and limitations of this study

Page 4, line 23: “Adjustment were made for age, sex and known comorbidities.” – what is meant by “known comorbidities”? -comorbidity that are associated with all-cause mortality?

This sentence has been removed and replaced by a sentence that included all variables adjusted for. We adjusted for comorbidities associated with all-cause mortality: age, sex, smoking, coronary heart disease, diabetes, hypertension, BMI and civil status.

Page 4: lines 8-10: “This is the first study to investigate the association between depressive symptoms and mortality in patients with AAA compared to a general population” vs lines 34-38: “The study focused on the association between depressive symptoms and mortality in patients with AAA.” do not test the same hypothesis and imply different analytic strategies.

Thank you for pointing this out. In retrospect, we see that these sentences do not belong in the short-list of “Strengths and limitations”. Hence, they have been replaced by a single statement related to strength and limitation of our study. Furthermore, we agree that our formulations of the aim were unclear. In the revised manuscript, the aim has been clarified both in the abstract and in the text.

Introduction:

Pages 5-6: “If depression influences the risk of AAA development, AAA patients suffering from depression may have higher mortality rates compared to AAA patients without depression.” The statement implies that if depression is associated with AAA development, then depression may act as an effect modifier in the AAA-mortality association. However, confounders are also associated with the exposure of interest by definition- so, the statement is inappropriate. Please rephrase and clarify the hypothesized interrelationships and pathways. Perhaps, you hypothesized that depression may act as an effect modifier in the AAA-mortality association through its association with inflammation?

Thank you for this comment. We have revised the sentence referred to by the reviewer. The reviewer does indeed raise some important and difficult questions regarding what is the effect modifier in view of hypothesized pathways.

Our initial hypothesis was that depression may trigger growth of the AAA - i.e. advance the severity of the disease, either as a promoter or a progressor (perhaps both). If this is the case, we could expect the adverse impact of depression on overall mortality rates to be more pronounced in individuals with an AAA compared to another subpopulation. In our study, the effect of depression was measured by comparing hazard of death (ratio, relative risk) between persons with and without depressive symptoms (HADS-D ≥ 8 vs. < 8). A similar adverse effect from depression might be present in other patient population. Hence, it is essential to compare the findings in the AAA group to those in a general population, which is more likely to reflect a general adverse effect of depression. Thus, to evaluate our hypothesis, we found it reasonable and correct to report HR values for depression in subgroups defined by the indicator variable for AAA. By definition, the AAA indicator variable, and not depression, would be the factor acting as an effect modifier on the association between depression and mortality.

Statistical analysis:

Page 9: "An interaction term between HADS and AAA was included in the model to evaluate whether the prognostic impact of depressive symptoms differed between persons with and without AAA."

Please specify that you mean multiplicative interaction term. Suggest you test both multiplicative and additive interactions. Also, some statements imply that your exposure of interest is AAA diagnosis, whereas depression is a potential effect modifier, while the sentence above implies that depression is your exposure variable of interest. Please be consistent and clear.

In accordance with the reviewer's comment, we have added a note regarding the use of a multiplicative interaction term (page 8 line 22). See also our comments above with respect to what factor that (technically speaking) is the effect modifier in the model.

In the revised manuscript, changes have been made to be more clear and consistent when reporting the results. In addition, the layout of Table 2 has been improved, to include more detailed descriptions on what factors we do report HR values for, and what subgroups.

Pages 9-10: "Predicted survival plots based on an adjusted Cox PH regression model, stratified for AAA and depressive symptoms, were constructed to allow for deviation from the PH assumption." How did you allow for deviation from PH assumption if the model was a PH model? Please note difference between terms 'time-varying covariate' and 'time-dependent effect'.

In the revised paper, we have replaced the phrase 'time-dependent covariates' with 'time-varying covariates (page 9, line 15-17) to distinguish better between age-dependent covariates and an age-dependent effect of the covariate (deviation from the PH assumption).

In view of the age-dependent effect seen in the unadjusted Kaplan-Meier plot, we constructed predicted survival plots based on the Cox-PH regression models, stratified on HADS-D and AAA to allow for deviation from the PH assumption. In the revised version of paper, we have moved the description of age-specific effects to the last paragraph in the result section (page 14-15)

Page 10: "Attained age was used as time scale both in the Kaplan-Meier survival analyses and in the Cox PH regression model." Survival function is hard to interpret when age is used as a time-scale and

hence not recommended. You could plot hazard function instead as it remains interpretable in terms of the risk experienced by people of a given age.

We agree with the reviewer. As recommended, we now present the expected proportions of death by attained age (failure plots) rather than regular survival curves (Figure 1 and Figure 2). Moreover, the description of the results in the text has been changed accordingly. We have also added a summary measure of survival time, i.e. median age at death, for subgroups that are compared, in the result section (page 13).

Results:

Page 11: provide a brief summary of descriptive statistics included in Table 1.

Changes have been made accordingly.

Discussion:

Page 15: "Previously reported mortality rates in AAA patients have most often been restricted to patients treated for AAA. [26, 28] In contrast, our study included both untreated and treated AAA patients." Could you adjust for the treatment?

Unfortunately, we did not have verified information about surgical treatment for AAA. We have added a sentence regarding this on page 16 line 23-24.

Reviewer: 2

Dr. Guy Martin, Imperial College London

Comments to the Author:

General Comments:

This prospective population-based study sought to identify the impact of depressive symptoms on mortality in patients with AAA. A sub-group of AAA patients (583, 1.6%) was identified from a larger study group.

On the whole it is a well written and constructed analysis from an established longitudinal study population with good data quality and completeness. The approach to analysis is well understood and has been extensively published by the study group in the past. The discussion and explanation of the results in the context of the wider literature is particularly well constructed.

Thank you for your thorough review of our manuscript and kind comments, we have replied to your questions below.

Generally, in the results, I find the narrative a little lost. The narrative is generally arguing that depression in AAA leads to a worse prognosis, yet the key result in Table 2 is that depression in those with AAA has no impact on mortality (HR 1.12, 0.88-1.51). I think the overall messaging of the result section does not fit with this and therefore should be revised to less strongly support the link between depression and outcomes in AAA. In fact, this is neatly highlighted in the discussion which is well written and framed; the results presented and how they are described in both the abstract and main text do not match that which is later discussed.

We do understand the confusion. We have made a major revision to the results section. Moreover, we distinguish better between what was our initial hypothesis (depression worsen the prognosis more in AAA patients compared to a general population) and the final results from our analyses (the opposite; impact of depression turned out to be weaker, not stronger in individuals with AAA).

Broadly speaking the overall message here is that of the single result that depression does not appear impact mortality in AAA that must be presented within the limitations of the statistical analysis. The remainder of the results presented merely demonstrate what is already well established in the literature (but in a relatively small sample size here) that AAA patients have a generally higher mortality risk, that most likely related to excess cardiovascular disease risks and smoking.

Thank you for this comment. Even though it was not the main aim of our study, we do believe that it is of importance to report the substantially excess overall mortality that was observed in patients with AAA compared to a general population of the same age. As the reviewer points out, the observed higher crude mortality rates in AAA patients have been attributed to other coexisting factors. Hence, it is important to adjust for potential confounders, since many risk factors for AAA also influence the risk of death.

However, going through the literature, we have not been able to find any previous study that in a similar way has reported relative risks based on adjusted analyses, or shown expected proportion of deaths by age in comparison to a general population of the same age and same risk profile. AAA is a relatively rare disease, which limits the opportunity to study this disease in an unselected population. Access to data from large, population-based surveys is thus essential, as we were able to in the HUNT study.

Specific Comments:

Abstract - The results/conclusions presented in the abstract do not seem coherent. Increased mortality in those with a AAA diagnosis compared to those without in the general population is expected and is not a novel or relevant finding. No data is provided for the size or significance for the effect of depressive symptoms in young patients with AAA despite stating they have a “particularly poor prognosis”. Essentially, the only conclusion that can really be drawn from the results provided is that depressive symptoms have no impact on mortality in patients with AAA. The results/conclusions presented require further clarity based on this.

We agree that the abstract could be improved. For the comment regarding the relevance of presenting the increased overall mortality and AAA, we refer to our response above. In accordance with the comment from the reviewer, we have rewritten the abstract to better emphasize our main aim (impact from depression on mortality), and furthermore clarified our results and conclusion.

P5L10 - arguably it is almost certainly 100% mortality, but some people do survive untreated ruptures in the short term.

We have made changes in the text in accordance with the reviewer's comment.

P5L45 - I wonder whether depression is a true independent risk-factor for AAA, or in fact is confounded by its association to smoking which is also a highly relevant risk factor for AAA. This was addressed in the original references provided which caveat the limitations of the statistical models used in them, in particular smoking being treated as a time-dependent variable. I think caveats such as this need to be addressed in framing the underlying hypothesis presented linking AAA mortality to

depression. Is it also that depressed people are less likely to engage with medical services and therefore not attend AAA screening? This should also be highlighted.

The reviewer raise some relevant issues, although not all are easy to comply with or adjust for. To avoid potential misunderstandings due to imprecise formulations, we have rephrased the sentence regarding depression being a risk factor for AAA (page 4 line 21-22). Smoking is an important and potential confounding factor both in our previous publication focusing on depression and risk of AAA and in the current manuscript. We were able to adjust for smoking status (never, previous, current), as well as other potential confounding factors. We agree that the discussion regarding confounding factors could be improved. In the revised version of the manuscript, we have extended the discussion regarding adjustment for confounding. In addition, we have added sentences in the limitations to take into account the use of self-report variables, including smoking, and the potential of unknown residual confounding. The limitation section includes a sentence about potential recruitment bias as a result of lower attendance in population-studies by individuals with severe depression.

P9L20 - there is no discussion of how the risk of collinearity of data has been assessed and mitigated (e.g., smoking with depression), and how this together with the number of AAA events may lead to over-fitting of the model. This should be discussed/accounted for.

Thank you for the comment. In general, it is difficult to provide a measure of collinearity for a log-linear model with categorical covariates. However, in the revised version of the manuscript, we have included a sentence in the methods (page 9 line 4-5) and in the results, to show how we have tried to account for this issue. There was no single factor that alone explained the reduction in HR value, neither in the AAA group or in the general population, or lead to a substantial increase in impreciseness of the relative risk estimates.

P11L20 - I think the key results discussed here should be presented in the text, rather than just referenced in a table (e.g., the adjusted HR for death in the AAA group, impact of younger age on prognostic impact of AAA). This should be done throughout the results section in order to make it easier for the reader to follow and understand.

We agree. In accordance with the suggestion from the reviewer, we now report the HR values for HADS-D and AAA in the result section, rather than just referring to Table 2. (page 11-12, line 18-21, line 10-12). We have also added some descriptive measures for subgroups that are compared.

P16L47 - the authors are correct to mention the potential risk of heritability on AAA in young patients. It would be sensible to specifically reference the connective tissue disorder cohort here who may be responsible for the weak association seen, but for whom data is not available.

Heritability is a strong risk factor for AAA. Indeed, some young patients with aneurysms have connective tissue disorders, such as Marfan, Loeys Dietz or Ehler Danlos. Furthermore, these patients have increased risk of premature death. We acknowledge that not being able to adjust for heritability is a limitation (as noted in the manuscript). However, we do not consider potential confounding from a connective tissue disorder cohort to be of concern in our present study, as we only included individuals diagnosed after the age of 60 years. We have included a sentence commenting on this issue in the discussion (page 17_line 11)

Figure 2A - the key problem with this figure, and therefore the relevance/significance of the findings is the small numbers at risk, particularly for patients with a AAA and HADS >8. Arguably the numbers are too small to make the plot justified.

In the submitted manuscript, we present ordinary Kaplan-Meier plots in Figure 1, whereas Figure 2 is predicted survival plots from a stratified Cox model (unadjusted, adjusted). In our opinion, the curves give an apprehensive description of the raw data that the overall HR values, as an average of all ages considered, are based on.

In accordance with a suggestion from reviewer no. 1, we have revised the Kaplan-Meier figures. In addition, we have rewritten the results section – and included useful information regarding median survival times (median age at death), to make the results from the plots more comprehensible. Furthermore, we have highlighted to problem with low numbers at risk, which preclude calculation of age-specific HR-values (page 15, line 7-9).

Reviewer: 1

Competing interests of Reviewer: None declared

Reviewer: 2

Competing interests of Reviewer: None declared

VERSION 2 – REVIEW

REVIEWER	Udumyan, Ruzan Örebro Universitet, Clinical Epidemiology and Biostatistics, School of Medical Sciences
REVIEW RETURNED	04-Jun-2021

GENERAL COMMENTS	Title: The prognostic impact of depressive symptoms on all-cause mortality in individuals with abdominal aortic aneurysm compared to the general population. A population-based prospective HUNT study in Norway Suggest small modification for more clarity: The prognostic impact of depressive symptoms on all-cause mortality in individuals with abdominal aortic aneurysm and in the general population. A population-based prospective HUNT study in Norway Abstract: P2 L7: “The aim was to examine the prognostic impact of depressive symptoms on all-cause mortality in individuals with AAA and compare with a general population of the same age and risk profile.” Suggest small modification for more clarity: The aim was to examine the prognostic impact of depressive symptoms on all-cause mortality in individuals with AAA and compare to that in a general population of the same age and risk profile. P2 L16: “There was no adverse effect of depressive symptoms in individuals with AAA (HR=1.15;95% CI 0.88-1.51),...” Suggest small modification: There was no clear association between depressive symptoms and all-cause mortality in individuals with AAA (HR=1.15; 95% CI 0.88-1.51),... P2 L21: “Depressive symptoms did not influence on mortality in the AAA group.”
---

	Suggest small modification: Depressive symptoms did not significantly influence mortality rate /or risk as you prefer if the outcome is rare/ in the AAA group. INTRODUCTION P5 L8: "...examined the prognostic impact of depression in individuals with AAA compared to a general population." Suggest changing to: ...examined the prognostic impact of depression in individuals with AAA compared to that in a general population. Statistical analysis P8 L18: "The Kaplan-Meier approach was applied to calculate the expected proportion of non-survivors (proportion of deaths) at different ages for AAA" Further to suggestion in the previous review, please plot hazard function instead as it remains interpretable in terms of the risk experienced by people of a given age: sts graph, by(variable) hazard P8 L24: "An interaction term between HADS-D and AAA (multiplicative model, using relative risk as measure of effect) was included in the Cox PH regression model to evaluate whether the prognostic impact of depressive symptoms differed between persons with and without AAA." Perhaps you mean 'multiplicative interaction term'. If yes, then please rephrase. For example: "A multiplicative interaction term was added to the Cox PH regression model to evaluate whether the association between depressive symptoms and all-cause mortality differed for persons with and without AAA." P9 L6: A backward stepwise procedure was applied to evaluate the individual contributions from single factors and to search for signs of potential overfitting of the model. Comment regarding overfitting: the statistical model is overfitted if it contains more parameters than can be justified by the data. For logistic and Cox models, the rule of thumb "one in ten rule" was suggested, i.e. minimum of 10 outcome events per coefficient in the model (paper by Vittinghoff E. "Relaxing the Rule of Ten Events per Variable in Logistic and Cox Regression"). P9 L10: Please add whether PH was satisfied for the exposure variable(s). P9 L10: Predicted survival plots based on the Cox PH regression model (unadjusted, adjusted) are also presented. What does this add? You may find this helpful: https://pclambert.net/software/standsurv/standardized_survival/ P9 L14: "Attained age was used as time scale both in the Kaplan-Meier survival analyses and in the Cox PH regression model." Please see the comment above regarding Kaplan-Meier analysis. Small suggestion: Cox proportional hazards models with attained age as the underlying time scale were fitted to estimate hazard ratios (HR) and 95% confidence intervals (CI) ...
--	---

	P11 L14: “A Kaplan Meier failure plot of the failure function, with expected proportion of deaths at different ages, for individuals with and without a diagnosed AAA, is shown in Figure 1A.” Please see the comment above. P11 L17: “The unadjusted hazard of death in individuals with AAA was more than three times the hazard of death in the general population (Table 2, HR 3.39, 95 % CI 3.01-3.78).” Please note that when you use attained age as a time-scale, then the model is age-adjusted rather than unadjusted. P12 L17: Age-specific effects Was PH assumption violated? If yes, then for which variable(s)? P 12 L23: The age-dependent pattern in the AAA group remained in the analyses adjusted for potential confounders, as illustrated in predicted survival plots based on the Cox-PH regression models, stratified on HADS-D and AAA to allow for deviation from the PH assumption (Figure 2). HR values are not calculated for stratification variables in the Cox PH model. Shall I conclude from here that the PH was violated for HADS-D and AAA? Although stratified Cox (stcox x1 x2, strata(x3)) is one of the ways to deal with PH violation, there are other solutions: e.g. modeling time-dependent effect for non-proportional predictors. Would suggest: 1) Running Cox PH model including the joint AAA and HADS_D variable 2) Testing PH assumption for the joint AAA and HADS_D variable 3) Modeling time-dependent effect for the joint AAA and HADS_D variable if PH assumption is violated. P25: Figure 2. Expected proportion of deaths in combined categories of AAA (no, yes) and HADS-D (<8 versus ≥8), based on a stratified Cox PH regression model. A. Unadjusted. B. Adjusted for age (time-scale), sex, BMI category, smoking, CHD, diabetes mellitus, civil status and hypertension. It might be better to present number of events and HRs for the joint AAA and HADS_D variable in a table.
--	--

REVIEWER	Martin, Guy Imperial College London, Department of Surgery & Cancer
REVIEW RETURNED	25-May-2021

GENERAL COMMENTS	On the whole I think the reviewer comments have been addressed and that the paper is now of far higher quality; the authors should be commended. Just one small typing error - P11L9 I think it should state “current smoker (57.5% vs. 35.5%)” in reference to risk factors being more prevalent in the AAA population.
--

VERSION 2 – AUTHOR RESPONSE

Response to Reviewers

Reviewer: 2

Dr. Guy Martin, Imperial College London

Comments to the Author:

General Comments:

On the whole I think the reviewer comments have been addressed and that the paper is now of far higher quality; the authors should be commended.

Just one small typing error - P11L9 I think it should state “current smoker (57.5% vs. 35.5%)” in reference to risk factors being more prevalent in the AAA population.

Thank you for pointing this out. Correction has been made (the numbers were indeed switched).

Reviewer: 1

Dr. Ruzan Udumyan, Örebro Universitet

Comments to the Author:

Title: The prognostic impact of depressive symptoms on all-cause mortality in individuals with abdominal aortic aneurysm compared to the general population. A population-based prospective HUNT study in Norway

Suggest small modification for more clarity: The prognostic impact of depressive symptoms on all-cause mortality in individuals with abdominal aortic aneurysm and in the general population. A population-based prospective HUNT study in Norway

Thank you for your suggestion. Changes have been made as suggested.

Abstract:

P2 L7: "The aim was to examine the prognostic impact of depressive symptoms on all-cause mortality in individuals with AAA and compare with a general population of the same age and risk profile."

Suggest small modification for more clarity: The aim was to examine the prognostic impact of depressive symptoms on all-cause mortality in individuals with AAA and compare to that in a general population of the same age and risk profile.

Changes have been made in accordance with suggestion from reviewer.

P2 L16: "There was no adverse effect of depressive symptoms in individuals with AAA (HR=1.15;95% CI 0.88-1.51),..."

Suggest small modification: There was no clear association between depressive symptoms and all-cause mortality in individuals with AAA (HR=1.15; 95% CI 0.88-1.51),...

Changes have been made in accordance with suggestion from reviewer.

P2 L21: "Depressive symptoms did not influence on mortality in the AAA group."

Suggest small modification: Depressive symptoms did not significantly influence mortality rate /or risk as you prefer if the outcome is rare/ in the AAA group.

We agree with the reviewer, and a reformulation has been made in accordance with the reviewer's comment.

INTRODUCTION

P5 L8: "...examined the prognostic impact of depression in individuals with AAA compared to a general population."

Suggest changing to: ...examined the prognostic impact of depression in individuals with AAA compared to that in a general population.

Reformulations have been made in view of suggestions from the reviewer (page 5, line 7-9)

Statistical analysis

P8 L18: “The Kaplan-Meier approach was applied to calculate the expected proportion of non-survivors (proportion of deaths) at different ages for AAA”

Further to suggestion in the previous review, please plot hazard function instead as it remains interpretable in terms of the risk experienced by people of a given age:

```
sts graph, by(variable) hazard
```

We believe it is important to keep the Kaplan-Meier failure plot since we report on descriptive measures extracted from these plots (expected no. of deaths by age and median age at death in each exposure group). Most readers of the journal are probably familiar with the interpretation of these standard plots.

We agree with the reviewer that it would be of interest to present plots of the hazard function, but then as additional information rather than replacing the Kaplan-Meier plots. As pointed out in the STATA manual (page 19, <https://www.stata.com/manuals/ststsgraph.pdf>), however, the smoothed hazard estimates are unreliable in the boundaries of range of data (i.e. youngest and oldest). In the small group with young AAA patients reporting depressive symptoms, the number of deaths were also low and occurred scattered and at unequal-spaced time intervals. Thus, we are unable to provide reliable estimate of the hazard in the complete age range based on the present data. We have already pointed out difficulties with providing age-specific HR estimates in the original manuscript.

In the revised version of our paper, and in accordance with the reviewer’s comments, we have added plots of the Nelson-Aalen cumulative hazard estimates as an alternative to the hazard function. These plots give a graphical display of the change in hazard with increasing age. The Nelson-Aalen estimate has been shown to be reliable even when data are sparse. The new plots appear in the renumbered Figure 1 (AAA) and Figure 2 (HADS-D in subgroups defined by AAA), respectively (right hand side). The text in the revised manuscript (methodological section, result section), are updated accordingly.

P8 L24: “An interaction term between HADS-D and AAA (multiplicative model, using relative risk as measure of effect) was included in the Cox PH regression model to evaluate whether the prognostic impact of depressive symptoms differed between persons with and without AAA.”

Perhaps you mean ‘multiplicative interaction term’. If yes, then please rephrase. For example: “A multiplicative interaction term was added to the Cox PH regression model to evaluate whether the association between depressive symptoms and all-cause mortality differed for persons with and without AAA.”

Reformulation has been made in accordance with suggestion from reviewer (page 9 line 1-3).

P9 L6: A backward stepwise procedure was applied to evaluate the individual contributions from single factors and to search for signs of potential overfitting of the model.

Comment regarding overfitting: the statistical model is overfitted if it contains more parameters than can be justified by the data. For logistic and Cox models, the rule of thumb “one in ten rule” was suggested, i.e. minimum of 10 outcome events per coefficient in the model (paper by Vittinghoff E. "Relaxing the Rule of Ten Events per Variable in Logistic and Cox Regression").

We commented on a potential problem with overfitting in view of previous comments from another reviewer. The reviewer talked about overfitting in combination with potential collinearity problems (rather than sample size). In the revised version of our manuscript, a more precise formulation on this issue is given (in view of initial remarks from reviewer). (Page 9, line 8)

P9 L10: Please add whether PH was satisfied for the exposure variable(s).

As reported in our manuscript, our main study variable (HADS-D in subgroups defined by AAA) shows signs of deviation from the PH assumption, i.e. not constant over time (age in our model). We report on this potential age-dependent effect in the last part of the result section. The observed finding of a potential age-dependent effect of depression in AAA patients is, in our opinion, quite interesting in view of our initial biological hypothesis, as further discussed in our paper.

In the revised version of our manuscript, we have added information on which adjustment factors that showed signs of deviation from the PH assumption (BMI-category), and report on results from analyses stratified on this factor (page 13, line 17-19)

P9 L10: Predicted survival plots based on the Cox PH regression model (unadjusted, adjusted) are also presented.

What does this add? You may find this

helpful: https://pclambert.net/software/standsurv/standardized_survival/

The purpose of showing the ‘unadjusted’ and adjusted survival plot, based on a Cox model stratified for HADS-D and AAA, was to illustrate extent of confounding in a model that allowed for an age-dependent effect of depression in subgroups defined by AAA. The unadjusted and adjusted HR values shown in Table 2 are calculated on basis of an ordinary Cox PH regression model, assuming constant effect over time (age). The overall HR values presented in Table 2, are probably not notably affected by the apparently more pronounced adverse effect of depression in young AAA patients.

As Lambert points out, the interpretation of the numerical values on the Y-axis in the adjusted survival curves will depend on the values of the covariates in the model. The distance between the adjusted survival curves, however, do not depend on covariate values (just a shift upwards or downwards). Nevertheless, we realize that a direct comparison with the unadjusted plot might be difficult. In the revised version of our paper, we have therefore removed the 'unadjusted' predicted survival plot (it doesn't add much information to the univariate Kaplan-Meier plot). The adjusted predicted survival plot based (Figure 2B in previous version, renumbered to Figure 4 in the revised version) is kept since this plot provides additional information with respect to time-related pattern in an adjusted model.

P9 L14: "Attained age was used as time scale both in the Kaplan-Meier survival analyses and in the Cox PH regression model."

Please see the comment above regarding Kaplan-Meier analysis.

Small suggestion: Cox proportional hazards models with attained age as the underlying time scale were fitted to estimate hazard ratios (HR) and 95% confidence intervals (CI)

As argued above, we believe it is important to keep the ordinary Kaplan-Meier plot, as these plots are easy to interpret and provide useful information. Attained age is used as time-scale both in the Cox PH regression model and in the univariate Kaplan-Meier analyses. However, we realize that the term 'covariate' is only defined for the Cox regression model.

In the revised version of our paper, reformulations have been made to correctly describe how we deal with the repeated measures of the exposure factors (page 9, line 22-24).

P11 L14: "A Kaplan Meier failure plot of the failure function, with expected proportion of deaths at different ages, for individuals with and without a diagnosed AAA, is shown in Figure 1A."

Please see the comment above.

As argued above, we believe it is important to keep the Kaplan-Meier plot. However, plots of the cumulative hazard function have been added as supplementary information.

P11 L17: "The unadjusted hazard of death in individuals with AAA was more than three times the hazard of death in the general population (Table 2, HR 3.39, 95 % CI 3.01-3.78)."

Please note that when you use attained age as a time-scale, then the model is age-adjusted rather than unadjusted.

We appreciate the comment. Changes have been made in accordance with the reviewer's comment in all parts of our manuscript, as well as in Table 2 (i.e we try to avoid the term 'unadjusted HR').

P12 L17: Age-specific effects

Was PH assumption violated? If yes, then for which variable(s)?

We have now added information on adjustment factors that showed signs of deviation for the PH assumption (BMI-category) (page 13 line 17-20). The heading of the last paragraph in the results section has been modified (Age-dependent effects, rather than age-specific effects). The observed age-dependent effect of our main study variables (HADS-D in subgroups of AAA) is already described in this paragraph.

P 12 L23: The age-dependent pattern in the AAA group remained in the analyses adjusted for potential confounders, as illustrated in predicted survival plots based on the Cox-PH regression models, stratified on HADS-D and AAA to allow for deviation from the PH assumption (Figure 2). HR values are not calculated for stratification variables in the Cox PH model.

Shall I conclude from here that the PH was violated for HADS-D and AAA? Although stratified Cox (stcox x1 x2, strata(x3)) is one of the ways to deal with PH violation, there are other solutions: e.g. modeling time-dependent effect for non-proportional predictors.

Would suggest:

- 1) Running Cox PH model including the joint AAA and HADS_D variable
- 2) Testing PH assumption for the joint AAA and HADS_D variable
- 3) Modeling time-dependent effect for the joint AAA and HADS_D variable if PH assumption is violated.

Yes, it is correct that our analyses have revealed a stronger adverse effect of depression in AAA patients <70 years of age (a deviation from the PH assumption). As highlighted in our paper, however, this finding needs to be confirmed in other studies. A graphical display of the age-dependent effect, based on a stratified Cox regression model adjusted for potential confounders, is illustrated in Figure 2 (renumbered to Figure 4 in the revised version). HR values are not calculated for the stratification variable in the Cox PH model.

The suggested analytic strategy (including an interaction with age to allow for age-specific effects, if we understand it correctly), would be possible in a parametric survival model, but not in the semi-parametric Cox PH regression model (time-scale not included as parameter in the model, impossible to add an interpretable interaction term with age). Even if we would have applied a parametric model, we would run into problems with estimation of age-specific HR values due to insufficient amount of data, particularly among the youngest.

See also our comment below regarding a single variable representing combined categories of AAA and HADS-D.

P25: Figure 2. Expected proportion of deaths in combined categories of AAA (no, yes) and HADS-D (<8 versus ≥8), based on a stratified Cox PH regression model. A. Unadjusted. B. Adjusted for age (time-scale), sex, BMI category, smoking, CHD, diabetes mellitus, civil status and hypertension.

It might be better to present number of events and HRs for the joint AAA and HADS_D variable in a table.

The number of deaths in combined categories of HADS-D and AAA are shown in Table 2, together with HR values for HADS-D in subgroups defined by AAA. In the revised version of our paper, we have added a plot that shows HR values based on an analysis of the joint AAA and HADS-D variable, using a common reference category (Figure 3).

VERSION 3 – REVIEW

REVIEWER	Udumyan, Ruzan Örebro Universitet, Clinical Epidemiology and Biostatistics, School of Medical Sciences
REVIEW RETURNED	27-Sep-2021

GENERAL COMMENTS	Comments for the manuscript: The prognostic impact of depressive symptoms on all-cause mortality in individuals with abdominal aortic aneurysm and in the general population. A population-based prospective HUNT study in Norway The paper has improved substantially. I have only some points to further polish the paper: Page 12, lines 11-19: “In the analyses adjusted for sex, smoking, coronary heart disease (CHD), diabetes, BMI, hypertension, and civil status (Table 2) there was no significant prognostic impact if /of ?/ depressive symptoms in individuals with AAA (HR 1.15, 95%CI 0.88-1.51), whereas a weak adverse effect remained in the general population (HR 1.23, 95%CI 1.17-1.30). The difference in the prognostic impact of depressive symptoms between the two groups did not reach statistical significance (p, test for interaction = 0.62). The factors contributing most to the reduced strength of the association with depressive symptoms were CHD, BMI and smoking. There was no single factor that explained the flattening of the risk estimate or led to very imprecise estimates.” If you had a larger subgroup (AAA=1 & HADS-D ≥8) with more events, then the HR might be statistically significant (potential type-II error?). Whether HRs of 1.15 or 1.23 are clinically significant or not is another question of course. In general, this might depend on the baseline risk and how serious the outcome is: that is, a 15% increase in a small baseline risk of a less than grave outcome might not be meaningful, but a 15% increase in incidence of a common
---

and/or potentially fatal outcome would be harder to discount. Hence, suggest some minor edits, for example:

In the analyses adjusted for attained age, sex, smoking, coronary heart disease (CHD), diabetes, BMI, hypertension, and civil status (Table 2) depressive symptoms were still associated with increased mortality risk in the general population (HR 1.23, 95%CI 1.17-1.30). Among individuals with AAA, the magnitude of the association was weaker and statistically non-significant (HR 1.15, 95%CI 0.88-1.51). The difference in these estimates did not reach statistical significance (P for interaction = 0.62). The factors contributing most to the reduced strength of the association with depressive symptoms were CHD, BMI and smoking.

1) Re the authors' comment in the response: "We believe it is important to keep the Kaplan-Meier failure plot since we report on descriptive measures extracted from these plots (expected no. of deaths by age and median age at death in each exposure group). Most readers of the journal are probably familiar with the interpretation of these standard plots."

How did you extract "median age at death" from KM plots? Was it done by identifying the age at which the yline(0.50) crosses the failure curve (sorry, but looks like so when comparing the text with the plots)?

Let's say yline(0.5) crosses the KM failure curve at age 70 years. This means 50% of subjects failed by the time 70 (with the unit of your time variable) after the -enter()- time. We can also say the median survival time is 70.

To get the median age at death one may use the following syntax:
univar _t if _d==1 & exposure ==

2) "If we stset the data so that the underlying time scale is attained age, then all the HRs will be adjusted for age in the Cox model". Please see slide 290.

https://biostat3.net/download/stata/lecture_day1to4_2019__3up.pdf
Thus, instead of writing "... model without adjustment for potential confounders," it would be more concise and informative to call that model "age-adjusted model". After all, age is an important confounder. Likewise in the table 2, the headings could be 'age-adjusted' and 'multivariable adjusted'.

FYI points:

3) Re: "The unadjusted Kaplan Meier failure plot, as well as the Nelson-Aalen cumulative hazard plot (Figure 2), indicated a more pronounced negative prognostic effect from depressive symptoms in AAA individuals younger than 70 years."

For descriptive purposes, it is also possible to estimate the average mortality rates within each age band (e.g. <70 and >70) using strate or stptime.

Please see pages 15-16:

<http://www.pauldickman.com/survival/stataintro.pdf>

and an example from Stata:

```
. webuse diet
```

```
Declare data to be survival-time data
```

```
. stset dox, origin(time dob) enter(time doe) id(id) scale(365.25)
```

```
fail(fail==1 3 13) noshow
```

	Calculate person-time and incidence rates per 1,000 person-years, tabulating in ten-year intervals, and excluding observations ≤ 40 or > 70 <pre>. stptime, per(1000) at(40(10)70) trim by(hienergy)</pre> 4) Re the comment in the response file: "The suggested analytic strategy (including an interaction with age to allow for age-specific effects, if we understand it correctly), would be possible in a parametric survival model, but not in the semi-parametric Cox PH regression model (time-scale not included as parameter in the model, impossible to add an interpretable interaction term with age)." Non-proportional hazards can be incorporated either by splitting (stsplit ...) or using tvc(). https://www.stata.com/manuals13/ststcox.pdf
--	---